# Linking Dietary Patterns to Autism Severity and Developmental Outcomes: A Correlational Study Using Food Frequency Questionnaires; The Childhood Autism Rating Scale, Second Edition; And Developmental Profile 3

**DOI:** 10.3390/biomedicines13051178

**Published:** 2025-05-12

**Authors:** Dimitar Marinov, Sevdzhihan Eyubova, Albena Toneva, Rositsa Chamova, Rozalina Braykova, Stanislava Hadzhieva, Ruzha Pancheva

**Affiliations:** 1Department of Hygiene and Epidemiology, Faculty of Public Health, Medical University of Varna, 55 Marin Drinov Str., 9000 Varna, Bulgaria; albena_t@abv.bg (A.T.); rchamova72@gmail.com (R.C.); rozalinabra@abv.bg (R.B.); slava79@abv.bg (S.H.); rouzha.pancheva@gmail.co (R.P.); 2Research Group Nutrilect, Department of Neuroscience, Research Institute, Medical University of Varna, 55 Marin Drinov Str., 9000 Varna, Bulgaria; s.eyubova@shu.bg; 3Department of Pedagogy and Management in Education, Faculty of Education, Konstantin Preslavsky University of Shumen, 115 Universitetska Str., 9700 Shumen, Bulgaria

**Keywords:** autism, ASD, food frequency questionnaire (FFQ), selective eating, nutritional deficiencies

## Abstract

**Background/Objectives**: Autism Spectrum Disorder (ASD) is characterized by social communication challenges and repetitive behaviors. Children with ASD often exhibit selective eating habits that may result in nutritional deficiencies and exacerbate developmental issues. While food frequency questionnaires (FFQs) are effective for dietary assessment, the links between food preferences, ASD severity, and developmental outcomes remain underexplored, particularly in Bulgaria. This study examines these relationships using validated tools. **Methods**: The present report constitutes a pilot, hypothesis-generating substudy of the broader NutriLect project. This substudy involved 49 children aged 2–12 years diagnosed with ASD. Dietary patterns were evaluated with a modified FFQ, while ASD severity and developmental profiles were assessed using the Childhood Autism Rating Scale, Second Edition (CARS-2) and the Developmental Profile 3 (DP-3). **Results**: Among 49 ASD children (mean age = 6.89 ± 2.15 years; 86% boys), 73.4% consumed grains/potatoes daily. Only 34.7% met combined fruit and vegetable recommendations. Only 36.7% met the recommendation for daily milk or other dairy product consumption. Fish was consumed at least twice weekly by only 22,4%. Furthermore, children with more severe autism were approximately 9.4 times more likely to consume grains daily (χ^2^ = 14.319, *p* = 0.006). Logistic regression analyses indicated that higher cognitive scores were strongly associated with lower grain (OR ≈ 0.044) and other dairy products consumption (OR ≈ 0.337), yet with greater fish intake (OR ≈ 3.317). In contrast, better communication skills corresponded to increased milk consumption (OR ≈ 5.76), and higher physical development scores predicted more frequent egg consumption (OR ≈ 4.40). **Conclusions**: The pronounced preference for grain and meat products, which are frequently ultra-processed, and avoidance of nutrient-dense foods in children with severe ASD symptoms underscore the need for tailored dietary interventions. These interventions must address sensory sensitivities, nutritional inadequacies, and the risks that selective nutrition can have on the nutritional status and development of the children.

## 1. Introduction

Autism Spectrum Disorder (ASD) is a complex neurodevelopmental condition characterized by persistent challenges in social communication, alongside restricted and repetitive patterns of behavior, interests, or activities. The global prevalence of ASD has risen significantly over the past few decades, necessitating a deeper understanding of various factors that may influence its severity as well as the overall health and neurological development of these children. One area of increasing interest within the scientific community is the potential influence of the neurological development of these children on their food choices and the potential consequences for their nutritional status and overall health.

Research has indicated that children with ASD often exhibit selective eating behaviors, characterized by strong food preferences, aversions, and a limited variety of accepted foods. Some cases may be as severe as models of single food intake, a liquids-only diet, or complete food refusal [1,2]. In particular, studies using retrospective dietary assessment methods such as food frequency questionnaires (FFQ) have suggested that children with ASD have significantly greater food refusal—especially regarding the intake of nutrient-dense foods like fresh fruits and vegetables—when compared to typically developing children (TDC) [3,4]. Different researchers also report less variety of food; a higher degree of inadequate/unbalanced dietary intake [4]; and significant differences in the consumption of dairy, legumes, and grains when comparing ASD children to TDC [5]. In fact, some studies report that more than half of ASD children completely avoid vegetables, and over 70% have an inadequate intake of fiber, vitamins (such as D and E), and minerals (such as calcium) [6]. Such behaviors can contribute to nutritional deficiencies that negatively influence neural development and exacerbate certain behavioral symptoms associated with ASD. Certain restrictive eating patterns, such as a lower intake of fiber and polyphenols, may also affect the gut microbiome, and the resulting shifts in host–microbe metabolism can influence neuroactive signaling and brain development. Supporting this, two urine-metabolomics studies—one using combined NMR + LC-MS platforms and another using GC-MS—found that children with ASD show altered urinary levels of indoxyl sulfate, N-α-acetyl-L-arginine, methyl-guanidine, and phenylacetylglutamine compared to children with normal neurodevelopment. The research also suggests higher levels of succinate and glycolate and lower levels of hippurate, 3-hydroxyphenylacetate, and related microbial phenolic conjugates than neurotypical peers [7,8]. This alteration may reflect microbiome changes associated with lower fiber and polyphenol intake. Furthermore, studies have reported a greater prevalence of children with low weight as well as high body fat and obesity among the ASD population, which also suggests that these extreme dietary behaviors can adversely affect overall nutritional status [9].

While dietary preferences and patterns among individuals with ASD have been widely researched, the extent to which these choices correlate with the severity of the disorder remains largely underexplored. There is limited research that combines dietary assessment methods with validated diagnostic tools like the Childhood Autism Rating Scale, Second Edition (CARS-2) and the Developmental Profile 3 (DP-3) to provide a multidimensional understanding of the relationship between diet and ASD. The available evidence is largely mixed, with some trials reporting no link between ASD severity and food preferences or food selectivity [10,11]. On the other hand, some studies suggest associations between food preferences and specific domains related to ASD severity, such as sensory over-responsivity, inflexible adherence to routines, and even social communication [12]. Moreover, cultural and regional differences in dietary habits remain understudied, with no investigations conducted in contexts such as Bulgaria.

This study aims to address these gaps by leveraging a modified FFQ alongside CARS-2 and DP-3 to assess correlations between food preferences, autism severity, and developmental profiles in children with ASD, hypothesizing that there is a significant relationship between the severity of ASD symptoms and specific food choices—with more severe cases being associated with more pronounced selective eating behaviors and stronger food preferences. Previous research utilizing FFQs in ASD populations has provided reliable insights into the nutritional inadequacies and selective eating behaviors common among these individuals. The FFQ has been shown to reliably assess the degree of food selectivity in ASD children, as well as their dietary preferences for specific foods [3,6,13]. Overall, this study aims to fill a critical gap in the literature by exploring the relationship between food choices and the severity of ASD symptoms using a cohort-based approach. By also employing validated tools such as CARS-2 and DP-3 to assess ASD severity, this substudy aims to provide insights into the correlations between nutrition and different aspects of neurodevelopment in ASD children and pave the way for more personalized and effective dietary interventions in Bulgaria for individuals with autism.

## 2. Materials and Methods

### 2.1. Study Overview and Participant Recruitment

The NutriLect Project is a randomized controlled trial approved by the Medical University of Varna’s Ethics Committee. It focuses on evaluating the nutritional status of children diagnosed with autism spectrum disorder (ASD) and cerebral palsy (CP). Participants were recruited through phone screenings and in-person interviews conducted at the Karin Dom Foundation in Varna and other rehabilitation centers serving families with special needs in Northeastern Bulgaria. Written informed consent was obtained from parents or legal guardians prior to participation. The present report constitutes a pilot, hypothesis-generating substudy of the broader NutriLect project; the sample was not powered for confirmatory inference but rather to yield preliminary effect-size estimates that will steer formal power calculations for the ongoing trial.

### 2.2. Eligibility Criteria

Children aged 2 to 12 years diagnosed with ASD by a qualified specialist were eligible for inclusion, provided their families intended to remain within the service area for the study duration. Participants were excluded if they had experienced acute medical conditions or severe infections within the 10 days preceding contact, if they presented with genetic syndromes or chronic conditions unrelated to ASD that might affect nutritional status, or if their parents or legal guardians did not fully understand the study terms or were unable to communicate in Bulgarian.

### 2.3. Data Collection and Assessment Tools

After obtaining consent, data were collected through a series of in-person assessments:Parents or primary caregivers completed a demographic questionnaire together with a modified FFQ specifically developed for the NutriLect project. Household annual income, highest parental education, and physician-diagnosed comorbidities were recorded via the demographic questionnaire. The FFQ drew on a food checklist created by the authors and consolidated individual items into nine major food groups: (1) grains and potatoes, (2) vegetables, (3) fruits, (4) milk, (5) other dairy products (excluding milk), (6) meat and poultry, (7) fish, (8) legumes, and (9) eggs. This grouping mirrors the categories used in the Bulgarian national dietary guidelines and keeps the questionnaire concise, thereby improving feasibility for caregivers of ASD children. The questionnaire was applied in the spring–summer–fall period. The majority of children had all their main courses at home or at the family-type institution centers where they reside. To reduce respondent burden, most items were consolidated into broad categories; for instance, every cereal-based food and potato preparation was merged into a single “grains/potatoes” group, so the questionnaire does not distinguish whole-grain staples from refined or highly processed products. No distinction between raw and cooked intake was required, because all listed items are customarily consumed after cooking or baking in Bulgaria. An FFQ was chosen because this method is generally well suited to dietary assessment in the ASD population. Reported intakes were evaluated against recommendations of the National Centre of Public Health and Analyses (NCPHA). The NCPHA advises that children aged 0–3, 3–7, and 7–18 years consume at least one daily serving from each of the following groups: grains or potatoes, vegetables, fruits, milk, and other dairy products. It further recommends a daily serving of at least one protein-rich food—meat, fish, legumes, or eggs. A widely accepted expert guideline in Bulgaria also specifies that fish should be offered at least twice per week as part of a healthy diet.Childhood Autism Rating Scale, Second Edition (CARS-2): Administered by a trained clinical psychologist, CARS-2 provided a quantitative measure of autism severity across 15 behavioral domains, including social interaction, communication, and sensory sensitivities.Developmental Profile 3 (DP-3): Conducted by trained psychologists, the DP-3 evaluated developmental milestones across five key domains: physical, adaptive, social-emotional, cognitive, and communication. This comprehensive assessment offered insights into the broader developmental context related to dietary habits and ASD severity.

All personal data were anonymized to ensure confidentiality.

### 2.4. Substudy Focus

This publication is a substudy of the NutriLect research project, a large-scale study in Bulgaria aimed at evaluating the impact of an individualized nutritional intervention—developed and supported by a multidisciplinary team—on the anthropometric outcomes and overall health of children with ASD and cerebral palsy (CP). The protocol for the intervention in children with CP is already officially published [14].

The current substudy centers on examining the dietary habits and symptom severity among children with ASD. Initially, 81 children (aged 2–12 years, diagnosed based on DSM-IV or DSM-5 criteria, depending on the timing of diagnosis) were included. However, participants with significant missing data in any of the key instruments (modified FFQ, CARS-2, or DP-3) were excluded, as incomplete responses would have compromised the reliability of the analysis. The study investigated whether dietary patterns met the official nutritional recommendations and explored potential links between autism severity (as measured by CARS-2 and DP-3) and specific dietary preferences, including the degree of food restriction.

### 2.5. Statistical Analysis

Data analysis was performed using SPSS Statistics Version 27. Descriptive statistics were used to characterize the sample, while cross-tabulations assessed relationships between categorical variables. Logistic regression models and Spearman’s rho tests examined the associations between dietary patterns and developmental domains. A *p*-value of <0.05 was considered statistically significant for all analyses. Anonymized data from FFQ, DP-3 and CARS-2 can be found as a Appendix A.

## 3. Results

Data from complete datasets of 49 children were analyzed. This subgroup had a mean age of 6.89 years (SD = 2.15 years), with 42 participants (86%) being boys. The demographic questionnaire suggested that 82% of households fell into the same middle-income bracket, and 76% of primary caregivers held at least a bachelor’s degree.

### 3.1. Comparison Between FFQ Results and NCPHA Recommendations

The data obtained from the FFQ were compared to the recommendations from NCPHA. An analysis of the FFQ data revealed that 73.4% of the children consumed grains and/or potatoes daily. However, only 53.0% consumed fruits and 48.9% consumed vegetables on a daily basis, with just 17 children (34.7%) meeting the NCPHA recommendations for both fruits and vegetables. Daily consumption of milk was reported by 24.5% of the children, and only 22.4% consumed other dairy products (such as cheese or yogurt) daily. Cross-tabulation indicated that 31 children (63.3%) did not include either milk or other dairy products in their everyday menus. Only four children (8.1%) included all four key food groups (grains/potatoes, fruits, vegetables, milk, and other dairy products) on a daily basis. Regarding fish consumption, 46.9% of the children had once-weekly consumption, while only 22.4% consumed fish twice a week. A detailed breakdown of the percentage of children meeting these nutritional criteria is presented in Table 1.

Our frequency analysis also showed that 45.1% of the children consumed eggs at least twice a week, and 45.1% consumed legumes at least twice a week. Furthermore, the frequency analysis suggested that 93.8% consumed meat or meat products at least twice a week, and 79.6% consumed meat or meat products three times a week.

### 3.2. Developmental Functioning and Food Frequency

Chi-square tests, logistic regression analyses, and Spearman’s rho were performed to compare the FFQ outcomes with the DP-3-based developmental functioning scores of the ASD children. Developmental functioning scores were determined for each DP-3 domain (physical, adaptive, social–emotional, cognitive, and communication) and categorized according to the standard scoring system (<70: Delayed; 70–84: Below Average; 85–115: Average; 116–130: Above Average; >130: Well Above Average).

An ordinal logistic regression was conducted to examine the association between DP-3 scores and different food groups. The results indicated a statistically significant inverse relationship between the cognitive scores of DP-3 and grain consumption. Specifically, for each one-unit increase in the DP-3 cognitive score, the log odds of being in a higher category of grain consumption decreased by 3.125 (β = −3.125, SE = 0.97, Wald = 10.382, *p* = 0.001, 95% CI: −5.025 to −1.224). In terms of odds ratios, this corresponds to exp(−3.125) ≈ 0.044, suggesting that children with higher cognitive scores had approximately a 96% reduction in the odds of higher grain consumption. The results showed a statistically significant negative association between cognitive scores and other dairy products consumption. Specifically, for each one-unit increase in the DP-3 cognitive score, the log odds of being in a higher “other dairy products” consumption category decreased by 1.087 (β = −1.087, SE = 0.525, Wald χ^2^ = 4.275, *p* = 0.039, 95% CI: −2.116 to −0.057). When exponentiated, the coefficient corresponds to an odds ratio of exp(−1.087) ≈ 0.337. This suggests that higher cognitive development is associated with approximately a 66% decrease in the odds of consuming other dairy products more frequently.

In contrast, the DP-3 Communication score was positively associated with milk consumption. A one-unit increase in the communication score corresponded to an increase in the log odds of being in a higher milk consumption category by 1.751 (β = 1.751, SE = 0.808, Wald χ^2^ = 4.700, *p* = 0.030, 95% CI: 0.168 to 3.333). The odds ratio for this predictor, exp(1.751), is approximately 5.76, suggesting that ASD children with higher communication scores had higher odds of consuming milk more frequently by nearly sixfold. The results also indicated a statistically significant positive association between cognitive development and fish consumption. Specifically, for each one-unit increase in the DP-3 cognitive score, the log odds of being in a higher fish consumption category increased by 1.199 (β = 1.199, SE = 0.539, Wald χ^2^ = 4.95, *p* = 0.026, 95% CI: 0.143 to 2.256). When exponentiated, this coefficient corresponds to an odds ratio of exp(1.199) ≈ 3.317, suggesting that children with higher cognitive scores had over three times the odds of consuming fish more frequently compared to those with lower scores. The analysis also indicated that for each one-unit increase in the physical development score, the log odds of being in a higher egg consumption category increased by 1.481 (β = 1.481, SE = 0.541, Wald χ^2^ = 7.486, *p* = 0.006, 95% CI: 0.42 to 2.542). Exponentiating the coefficient, the odds ratio is exp(1.481) ≈ 4.40, suggesting that children with higher physical development scores are over four times as likely to be in a higher egg consumption category compared to those with lower scores. The results from the ordinal logistic regression analysis are displayed in Table 2.

Investigating the relationships between FFQ and DP-3 domains by estimating Spearman’s rho also suggested potential associations (Figure 1). Once again, the cognitive domain showed the greatest correlations with FFQ, specifically in regard to the consumption of grains and fish. More specifically, children with lower cognitive development appeared to have an increased frequency of consumption of grains (r = −0.478; *p* < 0.001) and reduced frequency of consumption of fish (r = 0.356; *p* = 0.012). A weak positive correlation was observed between fruit consumption and social–emotional development (r = 0.300, *p* = 0.036), suggesting potential avoidance of fruits amongst children with lower scores in the social–emotional DP-3 domain. Using Spearman’s rho, there were no statistically significant correlations between FFQ and physical, adaptive, or communication development, or FFQ and the social domain. The results from the Spearman’s rho analysis are displayed in Table 3.

### 3.3. Autism Severity (CARS-2) and Food Frequency

The aim was to assess if there is a correlation between the specific frequency of consumption of any of the tracked food groups and the severity of autism based on CARS-2 results (moderate or severe autism). Descriptive analysis of the data revealed several correlations between the frequency of consumption of grains and scores related to autism severity (Table 4). Notably, the CARS-2 results showed a significant positive relationship between the frequency of grain consumption and autism severity (χ^2^ = 14.319; *p* = 0.006). Statistical analysis showed Spearman’s rho= 0.435 (*p* = 0.002), indicating potentially increased preferences of the children with more severe ASD toward grains.

The correlation remained significant when comparing the severity of autism based on CARS-2 versus whether the child meets the NCPHA recommendations or not (χ^2^ = 9.112; *p* = 0.003). Moreover, the risk estimate of 9.429 (95% CI: 1.888–47.079) suggested that children with severe autism are approximately 9.4 times more likely to consume grains daily compared to children with a moderate form of autism (based on CARS-2 score).

When performing descriptive analysis, the results also suggested correlations between milk consumption and several scores. Notably, the CARS-2 score correlated with milk consumption. However, Superman’s rho had non-significant results, as the children with the most severe delays (highest scores in CARS-2) had either very high or very low milk consumption, without a specific trend. There were no other statistically significant correlations between autism severity based on CARS-2 ratings and FFQ results.

## 4. Discussion

This study aimed to explore the relationship between food preferences and the severity of ASD symptoms in children aged 2 to 12 years in Bulgaria. By utilizing validated tools such as the CARS-2 and DP-3, this research attempts to assess how dietary preferences correlate with various developmental domains in children with ASD. The findings revealed several noteworthy correlations that contribute to the understanding of dietary behaviors in relation to ASD severity. Because these data come from a **pilot** cohort, the effect-size signals should be viewed as provisional; nevertheless, they furnish an empirical basis for calculating the sample-size requirements of the ongoing full-scale NutriLect trial.

One of the most striking findings was the distinct pattern in grain consumption. Logistic regression analyses revealed that higher cognitive scores were strongly associated with lower grain intake: for each one-unit increase in the DP-3 cognitive score, the odds of being in a higher grain consumption category decreased by approximately 96% (OR ≈ 0.044). This was supported by Spearman’s rho, which showed that children with lower cognitive performance tended to consume grains more frequently (r = –0.478, *p* < 0.001). Moreover, the CARS-2 assessment indicated that children with more severe autism were about 9.4 times more likely to consume grains daily than those with milder symptoms. The texture, taste, and predictability of grain products might be more acceptable to these children, who often exhibit rigid eating patterns and a limited range of accepted foods. Grains, especially processed ones that are crunchy or have a specific mouthfeel, may provide sensory stimulation that aligns with their preferences. On the other hand, the increased incidence of not meeting the recommendations for grain consumption in children with better developmental scores may reflect their greater ability to tolerate dietary interventions, such as gluten-free diets. Gluten-free diets are quite commonly applied in children with ASD in Bulgaria despite the lack of scientific support for their benefits [15]. Parents of children with milder ASD symptoms may be more successful in implementing dietary modifications, believing they might alleviate symptoms, even though evidence is lacking. This selective eating behavior and specific preferences underscore the importance of addressing dietary habits in therapeutic interventions. Different grain products can vary significantly in terms of nutritional density, energy density, and fiber content. Therefore, professionals should encourage parents to focus on offering grains that are more nutritious and less energy-dense, such as whole grains like whole wheat bread, brown rice, oats, and quinoa. Whole grains provide more fiber, vitamins, and minerals compared to refined grains and can help maintain a healthy weight and nutritional status in these children.

The analyses of other dairy product consumption revealed a more nuanced picture. Overall, other dairy products’ consumption (including items such as cheese or yogurt, but not milk) was inversely associated with cognitive scores; each unit increase in the DP-3 cognitive score was linked to a 66% reduction in the odds of higher other dairy products consumption (OR ≈ 0.337). This may suggest a preference for the textures or flavors of dairy products (possibly sweetened dairy products and salty cheese) among children with more significant developmental challenges, or that those with better developmental scores are more likely to limit certain dairy products. A similar relationship was not found for plain milk. Nevertheless, this pattern is somewhat unexpected and warrants further investigation to understand the underlying factors influencing these preferences.

In contrast, when examining milk separately, a higher DP-3 communication score was associated with a nearly sixfold increase in the odds of frequent milk intake (OR ≈ 5.76). This dichotomy suggests that while children with lower cognitive function may tend to consume a broader range of dairy products, those with stronger communication skills appear more likely to include milk in their diets. Such differences underscore the importance of evaluating individual dairy products and their associations with specific developmental domains. Notably, although descriptive analyses hinted at a relationship between milk consumption and autism severity, the results were inconsistent; children with the most severe delays displayed highly variable milk intake, and Spearman’s correlation did not yield a clear trend. This variability suggests that factors beyond those measured here may influence milk consumption patterns in ASD. Fish consumption, a critical source of omega-3 fatty acids for neurological development, was positively associated with cognitive performance. Higher DP-3 cognitive scores corresponded to over three times the odds of consuming fish more frequently (OR ≈ 3.317), indicating that children with better cognitive abilities are more likely to include fish in their diets. Additionally, children with higher physical development scores were over four times as likely to consume eggs frequently (OR ≈ 4.40), which may reflect enhanced motor skills and a greater ability to manage a variety of foods. Fruit and vegetable intakes were also generally low, with only 53.0% and 48.9% of children consuming fruits and vegetables daily. A weak positive correlation was noted between fruit consumption and social–emotional development (r = 0.300, *p* = 0.036), suggesting that children with more advanced social–emotional skills might be somewhat more inclined to eat fruits. Fruits and vegetables provide vital vitamins and antioxidants necessary for general health and development. The insufficient consumption of these nutrient-dense foods in children with severe developmental delays raises concerns about potential nutritional deficiencies that could further exacerbate developmental challenges.

The correlations observed in this study highlight the complex interplay between dietary habits and developmental outcomes in children with ASD. While the data suggest associations between food preferences and ASD severity, it is crucial to acknowledge that correlation does not imply causation. The selective eating behaviors observed may be both a consequence of ASD symptoms and a contributing factor to the exacerbation of these symptoms due to potential nutritional deficiencies.

### 4.1. Comparison with Previous Research

Previous research has yielded mixed results. One trial reported that ASD with restrictive food intake and ASD without had similar CARS-2 scores [10]. Another trial reported that food selectivity was positively related to parent-reported autism symptoms but unrelated to autism severity or linguistic and cognitive abilities as measured by professionals [11]. In contrast, a previous study suggested that sensory over-responsivity and inflexible adherence to routines or rituals that are part of the restricted and repetitive behaviors (RRB) criterion for ASD may underlie comorbid food selectivity in ASD. In addition, as meals often have social facets, food selectivity might also be related to deficits in the social communication domain [12]. The researchers reported that the group with food selectivity had higher scores in SCQ-RSI and communication subscale scores, reflecting more severe ASD symptoms for this group.

Our study aligns with the notion that specific ASD traits, such as sensory sensitivities and rigidity in behavior, may influence dietary preferences. The preference for grains and frequent avoidance of fish and fruits observed in children with more severe ASD symptoms supports the idea that food selectivity is intertwined with core ASD characteristics.

### 4.2. Strengths and Limitations

The strength of the current research lies in employing DP-3 and CARS-2, which are two well-established tools used to assess ASD severity across multiple domains. By utilizing both CARS-2 and DP-3 in this study, this substudy aims to provide a nuanced understanding of how food choices relate to ASD severity across multiple dimensions of development. These tools allow for the assessment of ASD severity not only in terms of core symptoms but also in relation to broader developmental delays or deviations that may influence or be influenced by dietary patterns.

Several limitations should be acknowledged. This study lacked a control group of typically developing children (TDC), which limits the ability to compare the dietary patterns of children with ASD to those without the disorder. Additionally, the sample size was relatively small, and the data were collected from a single region in Bulgaria, which may affect the generalizability of the findings.

The reliance on parent-reported data through the Food Frequency Questionnaire (FFQ) may introduce reporting biases. Parents may overestimate or underestimate their child’s food intake due to social desirability or recall inaccuracies. Moreover, the FFQ, while useful for assessing general dietary patterns, does not provide detailed information on portion sizes or nutrient intake, which could offer more precise insights into nutritional status. To keep the questionnaire manageable for caregivers, individual foods were grouped into broad categories. A limitation that resulted from this is that there is no differentiation between subtypes, such as whole-grain and refined-grain products. No formal reliability or validity testing was performed in regard to our modified FFQ because, to our knowledge, no food-related questionnaire of any kind has yet been validated in Bulgarian for ASD populations, and designing and validating one lay outside the scope of this substudy. Because this is a pilot-scale study, we refrained from adding socio-economic status, parental education, or current dietary interventions as covariates so as not to over-fit the regression models. More than 80% of participating households fell into the same middle-income bracket, which reduces—though does not eliminate—the likelihood of SES-related confounding. We acknowledge that some residual bias may remain and will include these variables in the fully powered analyses planned for subsequent NutriLect publications.

### 4.3. Implications for Dietary Interventions

The findings of this study have practical implications for dietary interventions aimed at children with ASD. Understanding that children with severe ASD symptoms are more likely to prefer grain products provides an opportunity to introduce more nutrient-dense grain options. Healthcare providers and caregivers can focus on incorporating whole grains rich in vitamins, minerals, and fiber to enhance the nutritional value of the preferred foods. Additionally, strategies to gradually introduce a wider variety of foods, including fruits and fish, could help mitigate potential nutritional deficiencies.

Addressing these sensory sensitivities and eating behaviors requires a multidisciplinary approach involving nutritionists, psychologists, and occupational therapists. Tailored interventions that consider individual preferences and sensory experiences can facilitate more flexible eating patterns and improve overall nutritional status.

### 4.4. Future Research

Future investigations should include a control cohort of typically developing children (TDC) to strengthen comparative analyses. Enlarging the sample and recruiting participants from a broader range of geographic regions and socioeconomic strata would further enhance the external validity of the findings. In addition, employing more granular dietary-assessment methods—such as multiple-pass 24 h recalls or prospective food diaries—would provide a more precise estimate of nutrient intake. Although these elements lay outside the scope of the present substudy, the larger NutriLect project is already collecting 3-day food diaries and is set to enroll age- and sex-matched TDC controls, with data collection and analysis expected to be completed by the end of 2025. The study will also include cluster analysis to identify potential subgroups of children with ASD based on both food preferences and symptom presentation. This method may help reveal distinctive dietary profiles and contribute to a more tailored understanding of nutritional and behavioral patterns within this population.

Future research should also consider investigating the underlying reasons for specific food preferences, particularly the unexpected patterns observed with dairy products other than milk. Qualitative research methods, such as interviews or observational studies, could explore the sensory and behavioral factors influencing dietary choices in children with ASD.

Longitudinal studies examining how dietary interventions impact ASD symptoms and developmental outcomes over time would provide valuable insights. Understanding whether improving nutritional status through targeted dietary changes can alleviate some ASD symptoms could have significant implications for treatment approaches.

## 5. Conclusions

This study revealed significant associations between food preferences and ASD severity among children in Bulgaria. Notably, children with lower cognitive scores and more severe autism symptoms were more likely to consume grains, while those with better cognitive and communication skills tended to have higher fish and milk intakes. These findings suggest that sensory sensitivities and selective eating behaviors may contribute to the observed dietary patterns. However, limitations such as a small, region-specific sample, reliance on parent-reported data, and the absence of a control group restrict the generalizability of the results. Future research with larger, more diverse populations and refined dietary assessments is needed to further clarify these relationships and inform targeted nutritional interventions for children with ASD.

## Figures and Tables

**Figure 1 biomedicines-13-01178-f001:**
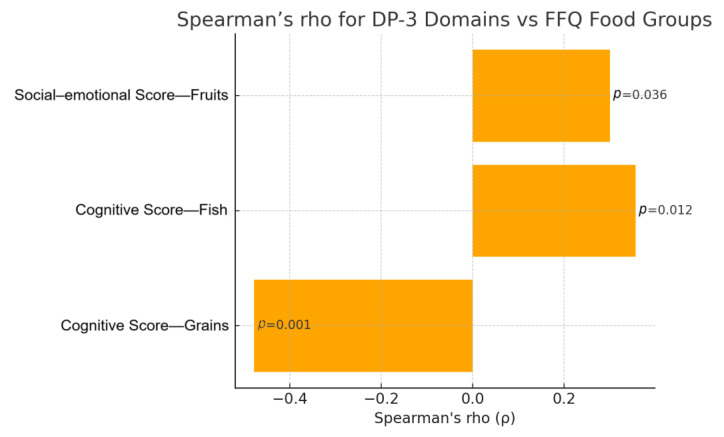
Spearman’s rank correlation between DP-3 domain scores and food frequency questionnaire outcomes.

**Table 1 biomedicines-13-01178-t001:** Percentage of ASD children meeting NCPHA recommendations.

Food Frequency Recommendations	Daily Consumption of Grains and/or Potatoes	Daily Consumption of Vegetables	Daily Consumption of Fruits	Either Daily Consumption of Milk or (and) Daily Consumption of Other Dairy Products	Twice Weekly Consumption of Fish
Percentage of ASD children meeting recommendations	73.4%	48.9%	53.0%	36.7%	22.4%

**Table 2 biomedicines-13-01178-t002:** Ordinal logistic regression analysis of food frequency in relation to DP-3 domains.

Predictor	Dependent Variable	β Estimate	Std. Error	Wald χ^2^	df	*p*-Value	95% Confidence Interval	Odds Ratio (exp(β))
DP-3 Cognitive Score	Grain Consumption Frequency (FFQ)	−3.125	0.97	10.382	1	0.001	−5.025 to −1.224	0.044
	Other Dairy Products Consumption Frequency (FFQ)	−1.087	0.525	4.275	1	0.039	−2.116 to −0.057	0.337
	Fish Consumption Frequency (FFQ)	1.199	0.539	4.95	1	0.026	0.143 to 2.256	3.317
DP-3 Physical Score	Egg Consumption Frequency (FFQ)	1.481	0.541	7.486	1	0.006	0.42 to 2.542	4.4
DP-3 Communication Score	Milk Consumption Frequency (FFQ)	1.751	0.808	4.7	1	0.03	0.168 to 3.333	5.76

**Table 3 biomedicines-13-01178-t003:** Spearman’s rho analysis on the relationship between different DP-3 domain scores (grades) and FFQ results.

DP-3 Domain	Food Group	Spearman’s Rho (ρ)	*p*-Value	Interpretation
Cognitive development	Grain consumption	−0.478	0.001	Lower cognitive scores were associated with higher consumption of grains, indicating potential preference.
	Fish consumption	0.356	0.012	Lower cognitive scores were associated with lower consumption of fish, indicating potential avoidance.
Social–emotional development	Fruit consumption	0.3	0.036	Lower social–emotional scores were associated with lower consumption of fruits.

**Table 4 biomedicines-13-01178-t004:** Frequency of grain consumption depending on CARS-2 results (moderate or severe autism).

N Days per Week with Grains	% of Children with Moderate ASD Based on CARS-2	% of Children with Severe ASD Based on CARS-2
2 or less	2.00%	2.00%
3	4.10%	6.10%
4	4.10%	0.00%
5	2.00%	6.10%
6	6.10%	67.30%

## Data Availability

The datasets presented in this article are not readily available because the data are part of an ongoing study. Requests to access the datasets should be directed to the corresponding author.

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
