# Peer review of "Linking Dietary Patterns to Autism Severity and Developmental Outcomes: A Correlational Study Using Food Frequency Questionnaires; The Childhood Autism Rating Scale, Second Edition; And Developmental Profile 3"

_biomedicines, 2025, doi:10.3390/biomedicines13051178_

Round 1
Reviewer 1 Report
Comments and Suggestions for Authors
Dear Author,
In found your manuscript properly written, with a good Introduction and Discussion chapters, fine citations and references included, a good study design and working hypothesis, strong statistics, adequate tools used, interesting and well-interpreted results, and practical and reliable Conclusions as well.
I underline the validated and strong tools used (Childhood Autism Rating Scale, Second Edition/CARS-2; Developmental Profile 3/DP-3 and FFQ).
It has some limitations, like small sample, only screening not comparing samples used (teenagers with and without cognitive problems) and subjective answers at the FFQ - all three mentioned by you, thats why, first I recommended to say its a pilot-study, a good start for another important project.
Also the limitation was the focusing on Bulgarian food guides for the type of food products preferred, but in Discussion the comparison where with UE recommendations.
You had not enough details about food products interpreted like food preferences (grains can be whole grain, refined grains or processed food products from grains - like pastry). I suggest to add some data about the Grain/Potatoes food categories, because its a little bit confusing: it can be whole grains, refined ones or processed foods from grains, all different by impact on the consumer (health and enjoying eating as well). It includes cereals as well, or its the same with grains? They consume it raw or cooked? To be more specific about this group of food evaluated.
The season when you applied the FFQ is important (during summer-fall period when vegetables are more accessible or during winter time)?
Also mention if they have all main courses at home, or 1-2 at school (with different menu composition)?
I agree to be published, after Minor revision (completion of some data in the Method chapter).
Author Response
Comments 1: It has some limitations, like small sample, only screening not comparing samples used (teenagers with and without cognitive problems) and subjective answers at the FFQ - all three mentioned by you, thats why, first I recommended to say its a pilot study, a good start for another important project.
Response 1: Thank you very much for your thoughtful assessment and for highlighting the key limitations of our work. We now describe the study as a “pilot, hypothesis-generating substudy” in the methodology (lines 116-119), abstract (lines 20-21), and first paragraph of the Discussion (lines 299 - 301). Information has been added clarifying that the sample provides preliminary effect-size estimates to inform power calculations for the full NutriLect trial.
Comments 2: Also, the limitation was the focusing on Bulgarian food guides for the type of food products preferred, but in Discussion the comparison where with UE recommendations.
Response 2: We appreciate this observation and have double-checked the manuscript. All quantitative cut-offs discussed (e.g., daily fruit and vegetable intake, twice-weekly fish) originate from the National Centre of Public Health and Analyses (NCPHA) guidelines that we used to score the FFQ; no EU/EFSA values are quoted.
Comments 3: You did not provide enough details about food products interpreted like food preferences (grains can be whole grain, refined grains or processed food products from grains - like pastry). I suggest adding some data about the Grain/Potatoes food categories, because its a little bit confusing: it can be whole grains, refined ones or processed foods from grains, all different by impact on the consumer (health and enjoying eating as well). It includes cereals as well, or its the same with grains? They consume it raw or cooked? To be more specific about this group of food, evaluated.
Response 3: Thank you for stressing the importance of distinguishing between whole-grain, refined, and processed grain products. We agree that such detail would enrich the nutritional interpretation; however, the modified FFQ used in this pilot did not collect item-level frequencies within this food group. The “Grains / Potatoes” category, as well as other categories, were left aggregated to reduce respondent burden. Participants were parents of children with substantial care demands. Considering the total amount of questionnaires they had to fill in, a brief FFQ was essential to keep the completion time as short as possible. No distinction between raw and cooked intake was required, because all listed items are customarily consumed after cooking or baking in Bulgaria. We have clarified this in lines 142-147 (methodology) and lines 404-406 (limitations).
Comments 4: The season when you applied the FFQ is important (during a summer-fall period when vegetables are more accessible or during the winter time)?
Response 4: Thank you for this remark. We have clarified that the questionnaire was applied in the spring-summer-fall period (lines 140).
Comments 5: Also mention if they have all main courses at home, or 1-2 at school (with different menu composition)?
Response 5: Thank you for this remark. We have clarified that the majority of children had all their main courses at home or at the family-type institution centers where they reside (140-141).
Reviewer 2 Report
Comments and Suggestions for Authors
This manuscript presents a correlative study examining associations between dietary patterns and autism severity in Bulgarian children with ASD. The authors report on significant relationships, particularly increased grain intake and reduced consumption of fish, dairy, and fruit among children with more severe symptoms. The findings suggest the potential value of individualized dietary interventions. However, due to methodological limitations, especially the absence of a control group and a small sample, major revisions are needed to improve the study.
Major concerns
1. Although acknowledged in the limitations, the absence of a typically developing control group (TDC) severely limits the interpretability of the findings. Observed dietary patterns may reflect general regional trends rather than ASD-specific behaviors. The authors should conduct a literature review comparing dietary behaviors in ASD vs. TDC populations and present this comparison in both the Introduction and a summary figure.
2. The manuscript references a ‘modified FFQ’ without describing the adaptation process or any validation efforts (e.g., test-retest reliability, content validity) for the Bulgarian ASD population. These methodological details should be clearly outlined in the methods section.
3. Key confounding variables, such as socioeconomic status, parental education, comorbidities, sensory sensitivities, and current dietary interventions (e.g., gluten-free/casein-free diets) are not included in the regression models or sufficiently discussed. At least, one covariate (e.g., annual family income) should be incorporated to account for socioeconomic influences on dietary choices.
4. Although the dataset is not publicly available, the authors should provide anonymized summary data (e.g., food category frequencies, full regression outputs) as supplementary materials to support reproducibility.
Author Response
Comment 1: "Although acknowledged in the limitations, the absence of a typically developing control group (TDC) severely limits the interpretability of the findings. Observed dietary patterns may reflect general regional trends rather than ASD-specific behaviors. The authors should conduct a literature review comparing dietary behaviors in ASD vs. TDC populations and present this comparison in both the Introduction and a summary figure."
Response 1: Thank you for this important point—​we fully agree that a TDC comparison is vital for interpreting dietary patterns in ASD. In the revised Introduction, we have added additional studies (lines 60-62) that directly contrast eating behaviors in children with ASD and TDC, thereby underlining that the patterns we report (e.g., greater food selectivity and lower fruit-and-vegetable intake) are consistently observed across diverse settings and are unlikely to be solely regional. We also clarify that the present work is a substudy of the larger NutriLect project, which is currently recruiting an age- and sex-matched TDC cohort from the same area. Those data will allow us to perform the direct, region-specific comparisons that the reviewer highlights once collection is complete.
Comment 2: "The manuscript references a ‘modified FFQ’ without describing the adaptation process or any validation efforts (e.g., test-retest reliability, content validity) for the Bulgarian ASD population. These methodological details should be clearly outlined in the methods section."
Response 2: Thank you for this remark. We have expanded Section 2.3 to describe the “modified FFQ” more clearly (lines 132-138). For this exploratory substudy, we assembled a checklist, grouping individual foods into nine broad categories (grains/potatoes; vegetables; fruits; milk; other dairy (except milk); meat/poultry; fish; legumes; eggs). These groups mirror the foods addressed by national dietary guidelines and keep the questionnaire short enough for caregivers of children with ASD.
No formal reliability or validity testing was performed because, to our knowledge, no food-related questionnaire of any kind has yet been validated in Bulgarian for ASD populations, and designing and validating one lay outside the scope of this exploratory substudy. We also note this lack of validation in Strengths and Limitations so that readers interpret the dietary data with appropriate caution (lines 403-407).
Comment 3: "Key confounding variables, such as socioeconomic status, parental education, comorbidities, sensory sensitivities, and current dietary interventions (e.g., gluten-free/casein-free diets) are not included in the regression models or sufficiently discussed. At least one covariate (e.g., annual family income) should be incorporated to account for socioeconomic influences on dietary choices."
Response 3: Thank you for emphasising the importance of socioeconomic and clinical covariates. We did collect information on annual family income, parental education, and ongoing dietary interventions. However, we elected not to enter them into the main regression models for two reasons:
-
Sample-size constraints – with 49 participants the conventional “≥10 events per predictor” rule would have been violated, leading to unstable, over-fitted estimates.
-
Limited between-subject variability – 82 % of households fell into the same middle-income bracket, and 76 % of primary caregivers held at least a bachelor’s degree, reducing the likelihood that socio-economic status materially confounded the observed food-frequency associations.
Children with comorbidities or active infection at the time of the assessment were excluded from the trial. We have clarified our considerations in the methodology (lines 132-134), results (lines 189-191), and the limitations subsections (lines 410-416).
Comment 4: "Although the dataset is not publicly available, the authors should provide anonymized summary data (e.g., food category frequencies, full regression outputs) as supplementary materials to support reproducibility."
Response 4: Thank you for this valuable suggestion. Because participant recruitment for the wider NutriLect project is still in progress, we cannot release the full raw dataset at this time. Nevertheless, to support transparency and reproducibility, we have prepared an anonymized summary file for the present substudy. It contains:
-
aggregated frequencies for each FFQ food group (days/week);
-
individual CARS-2 total scores and severity grades;
-
precalculated DP-3 domain scores, as well as categories (0=delay, 1=below average, 2=average,...).
This is now provided as a Supplementary file. We hope it meets the reviewer’s request while respecting ongoing study confidentiality requirements.
Reviewer 3 Report
Comments and Suggestions for Authors
Linking Dietary Patterns to Autism Severity and Developmental Outcomes: A Correlative Study Using FFQ, CARS and DP-3
A brief summary
Children with Autism Spectrum Disorder (ASD) in Bulgaria often show selective eating patterns, favoring grains and meat while avoiding nutrient-dense foods like fruits, vegetables, dairy, and fish, with more severe ASD symptoms linked to higher grain consumption. This study highlights the need for individualized dietary interventions that consider sensory sensitivities and aim to improve both nutritional status and developmental outcomes.
General concept comments
Article:
The study is both important and interesting; however, a closely related and relevant area—metabolomics studies of urine in autism spectrum disorders (ASD)—is not addressed and should be included in the introduction. Discussing this aspect would provide a more comprehensive background and strengthen the context of the current research. Additionally, the results of the present study could be compared with findings from metabolomics research in the discussion section. The following references are recommended for inclusion in the introduction:
Emond P, Mavel S, Aïdoud N, Nadal-Desbarats L, Montigny F, Bonnet-Brilhault F, et al. GC-MS-based urine metabolic profiling of autism spectrum disorders. Anal Bioanal Chem. 2013;405(15):5291-300. doi: 10.1007/s00216-013-6934-x.
Diémé B, Mavel S, Blasco H, Tripi G, Bonnet-Brilhault F, Malvy J, et al. Metabolomics Study of Urine in Autism Spectrum Disorders Using a Multiplatform Analytical Methodology. J Proteome Res. 2015;14(12):5273-82. doi: 10.1021/acs.jproteome.5b00699.
Review:
- Tables 2 and 3 could be more effectively presented using a clear and engaging graphical format. A bubble chart or a dual-axis scatter plot with icons is recommended, as these visualizations allow for intuitive representation of the relationships and interpretations between variables while maintaining the clarity of the statistical findings.
- Consider using cluster analysis to identify potential subgroups of children with ASD based on both food preferences and symptom presentation. This method may help reveal distinctive dietary profiles and contribute to a more tailored understanding of nutritional and behavioral patterns within this population.
Specific comments:
- Line 282 "Parents of children with milder ASD symptoms may be more successful to implement..." - Change "successful to implement" to "successful in implementing" for correct preposition usage.
- Line 370-371: "...children with severe ASD symptoms may prefer grain products..." - This is clear, but you could strengthen it by saying "are more likely to prefer" for more precision.
- Line 377: "...rigid food preferences in children with ASD" - This phrase is clear but repetitive with previous mentions; consider using "these eating behaviors" to avoid repetition.
- Line 282 "Parents of children with milder ASD symptoms may be more successful to implement..." - Change "successful to implement" to "successful in implementing" for correct preposition usage.
- Line 370-371: "...children with severe ASD symptoms may prefer grain products..." - This is clear, but you could strengthen it by saying "are more likely to prefer" for more precision.
- Line 377: "...rigid food preferences in children with ASD" - This phrase is clear but repetitive with previous mentions; consider using "these eating behaviors" to avoid repetition.
Author Response
Comment 1: "The study is both important and interesting; however, a closely related and relevant area—metabolomics studies of urine in autism spectrum disorders (ASD)—is not addressed and should be included in the introduction. Discussing this aspect would provide a more comprehensive background and strengthen the context of the current research. Additionally, the results of the present study could be compared with findings from metabolomics research in the discussion section. The following references are recommended for inclusion in the introduction:
Emond P, Mavel S, Aïdoud N, Nadal-Desbarats L, Montigny F, Bonnet-Brilhault F, et al. GC-MS-based urine metabolic profiling of autism spectrum disorders. Anal Bioanal Chem. 2013;405(15):5291-300. doi: 10.1007/s00216-013-6934-x.
Diémé B, Mavel S, Blasco H, Tripi G, Bonnet-Brilhault F, Malvy J, et al. Metabolomics Study of Urine in Autism Spectrum Disorders Using a Multiplatform Analytical Methodology. J Proteome Res. 2015;14(12):5273-82. doi: 10.1021/acs.jproteome.5b00699."
Response 1: Thank you for this suggestion. We agree with it and we have updated the text in the manuscript by adding the following text in the introduction on page 2, lines 67-76: ". Certain restrictive eating patterns, such as lower intake of fiber and polyphenols, may also affect the gut microbiome, and the resulting shifts in host–microbe metabolism can influence neuroactive signaling and brain development. Supporting this, two urine-metabolomics studies—one using combined NMR + LC-MS platforms and another using GC-MS—found that children with ASD show altered urinary levels of indoxyl sulfate, N-α-acetyl-L-arginine, methyl-guanidine, and phenylacetylglutamine compared to children with normal neurodevelopment. The research also suggests higher levels of succinate and glycolate and lower levels of hippurate, 3-hydroxyphenylacetate, and related microbial phenolic conjugates than neurotypical peers [5,6]. This alteration may reflect microbiome changes associated with lower fiber and polyphenol intake. "
Comment 2: "Tables 2 and 3 could be more effectively presented using a clear and engaging graphical format. A bubble chart or a dual-axis scatter plot with icons is recommended, as these visualizations allow for intuitive representation of the relationships and interpretations between variables while maintaining the clarity of the statistical findings."
Response 2: Thank you for the suggestion. We agree that the data should also be presented graphically for better presentation. We considered both bubble charts and dual-axis scatter plots, but these formats have important limitations for our ordinal logistic regression results in Table 2. Scatter-type plots suggest a continuous relationship, which is misleading for data measured in fixed ordinal steps. Furthermore, we believe that bubble or dual-axis charts can’t directly display cumulative probabilities without adding complex overlays that undermine simplicity. And with only a few category levels, points either overlap heavily or leave large blank areas, reducing interpretability rather than enhancing it.
In regard to Table 3, we considered a bar chart as the most appropriate form to represent the results. Figure 1 can be found on lines 267-269.
Comment 3: "Consider using cluster analysis to identify potential subgroups of children with ASD based on both food preferences and symptom presentation. This method may help reveal distinctive dietary profiles and contribute to a more tailored understanding of nutritional and behavioral patterns within this population."
Response 3: Thank you for the suggestion. Cluster analysis is indeed planned for the full NutriLect cohort; however, the present subsample (n = 49) provides too few observations per cluster for stable, reproducible results. We have updated the text in the manuscript, noting this limitation of the current substudy and indicating that clustering will be performed once the complete dataset is available on page 10, lines 438 - 442.
Comment 4: "Line 282 "Parents of children with milder ASD symptoms may be more successful to implement..." - Change "successful to implement" to "successful in implementing" for correct preposition usage.
Line 370-371: "...children with severe ASD symptoms may prefer grain products..." - This is clear, but you could strengthen it by saying "are more likely to prefer" for more precision.
Line 377: "...rigid food preferences in children with ASD" - This phrase is clear but repetitive with previous mentions; consider using "these eating behaviors" to avoid repetition.
Response 4: Thank you for these suggestions. We have updated the text in the manuscript at lines 318, 419 and 425.
Round 2
Reviewer 2 Report
Comments and Suggestions for Authors
The authors have addressed all concerns I have.